# Understanding networks in rural Cambodian farming communities and how they influence antibiotic use: A mixed methods study

**Jane Mingjie Lim**[1], **Sokchea Huy**[2], **Ty Chhay**[2], **Borin Khieu**[3], **Li Yang Hsu**[1], **Clarence C. Tam**[1,4] *

1 Saw Swee Hock School of Public Health, National University of Singapore and National University Health System, Singapore, Singapore, 2 Centre for Livestock and Agriculture Development (CelAgrid), Phnom Penh, Cambodia, 3 General Department of Local Communities, Ministry of Environment, Phnom Penh, Cambodia, 4 London School of Hygiene & Tropical Medicine, London, England, United Kingdom

* clarence.tam@nus.edu.sg

**Data Availability Statement:** The minimal anonymized dataset is available at the National University of Singapore (ScholarBank@NUS Repository; https://doi.org/10.25540/51MK-KH4C).

## Abstract

Biosecurity and preventive animal health services in Cambodian smallholder backyard farming systems are often limited, leading to an over-reliance on antibiotics. However, data on factors influencing antibiotic use in these settings are lacking. We conducted a study in two rural Cambodian farming communities to investigate how social and contextual influences affect both human and animal antibiotic use behaviours. Data were collected in three phases: a baseline household census to enumerate village residents, a social network survey to understand village-level social ties, and in-depth interviews to elicit information about the influence of social ties on their decision-making processes. Primary outcome measures included knowledge, attitudes and practices surrounding antibiotic use, and awareness of issues relating to antibiotic resistance. Participants commonly accessed antibiotics or learned animal antibiotic use practices through village-level informal sources such as pharmacies or animal health workers. While most participants reported not using antibiotics for animal growth promotion or illness prevention, misconceptions surrounding both antibiotic effectiveness and resistance were common. Social networks capturing informal, work-related and health-related social ties showed that familial connections and geographic proximity were of primary importance for information sharing. Using exponential random graph models, we demonstrated that familial ties, and closer geographic and geodesic distance, were associated with similarity in overall antibiotic knowledge and attitudes. The informal private sector plays a major role in provision of antibiotics and antibiotic-related information in backyard farming communities, but such information is maintained within close social groups. This demonstrates the importance of engaging village-level informal sources in the provision of antibiotic-related information for both human and animal health, as well as in potential interventions to encourage appropriate antibiotic behaviours in lower-resourced settings.

**Funding:** This work was funded by the UHS-SPH Integrated Research Programme (USIRP) at the Saw Swee Hock School of Public Health, National University of Singapore (NUS) and the University of Health Sciences, Cambodia (UHS).

**Competing interests:** The authors have declared that no competing interests exist.

## Background

Smallholder backyard farming systems comprise a significant portion of Cambodia's gross domestic product and serve a substantial economic role in many rural Cambodian households—more than half of all households keep some poultry [1]. Biosecurity and preventive animal health services in these settings are limited, while antibiotics to treat both human and animal infections are widely available without prescription through formal and informal sellers. To ensure quality of feed and veterinary drugs, the Ministry of Agriculture, Fisheries, and Forestry and the General Directorate of Animal Health and Production jointly established regulation in 2018 requiring the registration of imported animal feed and veterinary drugs. However, there are limited data on antibiotic use in the agricultural sector and the pathways through which agricultural communities access antibiotics [2].

While village and animal health agents advise on livestock health for most backyard farming communities in Cambodia, unrestricted access to antibiotics for self-medication is common via pharmacies and other drug outlets [3,4] such as untrained pharmacists, animal feed stores and grocery stores. Further, the interconnectedness of humans and animals in communities with high concentrations of backyard farming can facilitate inappropriate antibiotic use and sharing of antibiotics between humans and animals. Understanding the scale and impact of these practices is essential, to design effective interventions addressing setting-specific inappropriate use of antibiotics [5].

To understand the flow of antibiotics and health-related information in smallholder farming operations, and how this influences human and animal antibiotic use behaviours, we conducted a mixed methods, social network study within two Cambodian farming communities.

## Methods

We recruited participants from two farming villages in Takeo, a primarily rural province in southern Cambodia bordering Vietnam (**Fig 1**). Each village comprised approximately 100 households. Data were collected in three phases: a baseline household census to enumerate village residents, a social network survey, and in-depth interviews with key informants in the network. Participants were reimbursed with a token of USD3 and USD8 for the household census and in-depth interviews respectively.

### Patient and public involvement

No members of the public were involved in the study. To inform the content of our study tools, we pilot tested the study materials with members of the public external to our research. Key findings will be disseminated to study respondents through the administration team at CelAgrid.

### Household census

We first conducted a photographic census (S1 Appendix) of adults residing in both villages to facilitate nomination of social ties in the subsequent network survey. A community leader accompanied the research team to enumerate occupied households. Households were eligible for inclusion if 1) they owned farm animals, and 2) farming was a main source of household income. In each eligible household, we invited all adults (aged $\geq$ 21 years) to participate. Individuals who provided verbal consent had their photograph taken using a smartphone. Photographs were immediately printed using portable wireless printers, labelled with a unique participant identifier embedded in a QR code, and placed in a photo album on a page labelled with a unique QR code for the household (**Fig 2**). Participating household coordinates were also recorded using global positioning system (GPS) receivers on electronic tablets.

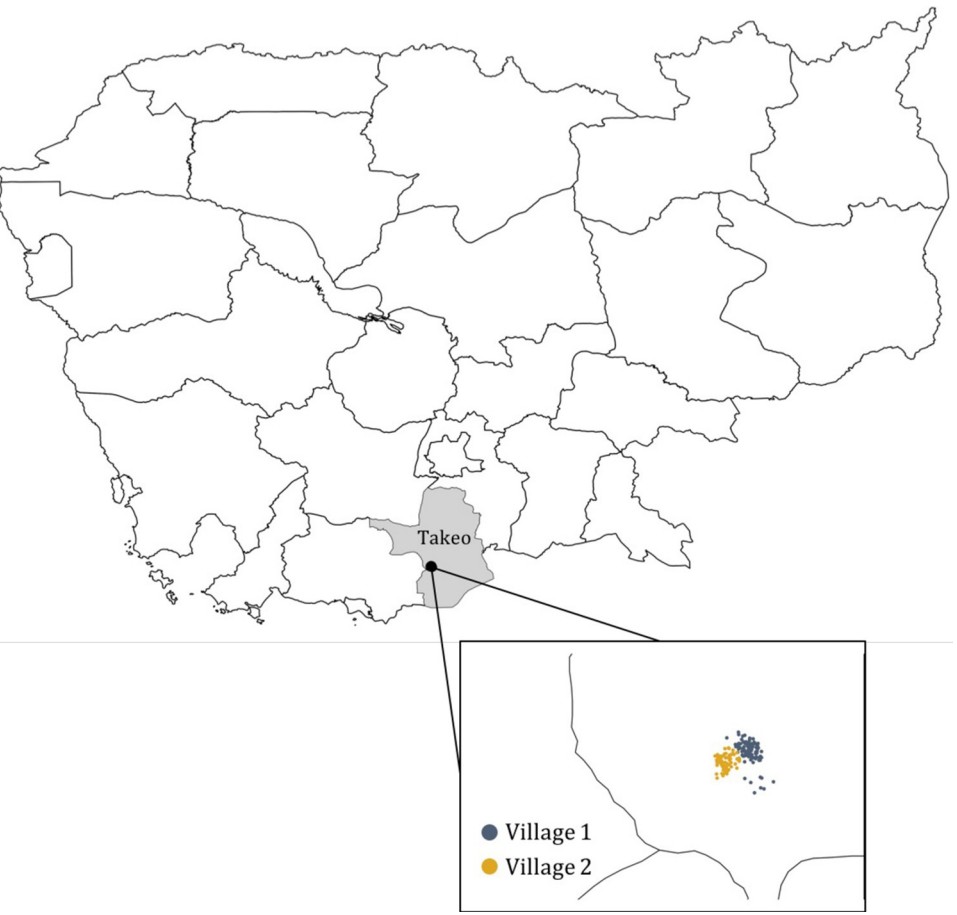

**Fig 1. Villages 1 and 2 in Takeo, Cambodia.** This map was created in R using the package rnaturalearth [6] using maps from Natural Earth (http://www.naturalearthdata.com/) where all maps are public domain.

### Social network survey

Following the photographic census, we returned to participating households to conduct a social network survey (S2 Appendix). In each household, we interviewed one adult using a standardised questionnaire. We collected information about participants' demographic and socioeconomic details, animal ownership, knowledge, attitudes and practices surrounding antibiotic use, and awareness of issues relating to antibiotic resistance. We also asked participants if they had leftover antibiotics at home at the time of interview. For those who did, we requested permission to take photographs of the antibiotics for verification.

Participants were then shown the photo albums compiled during the census and asked to nominate up to three key social ties across different domains of their life, including who they spent the most time talking to day-to-day (talking network), who they discussed work-related issues with (work-related network), who they obtained health advice from (health network), and who they consulted for animal health matters.

Participants' social tie nominations were recorded by scanning the relevant QR codes in the photo album identifying the household and individual. If the participant nominated a village resident not captured in our census, we added a unique QR code to the affiliated household and used that as a proxy for that nomination. The network survey was conducted by trained field staff in Khmer and took approximately 20–30 minutes per household. Data were collected

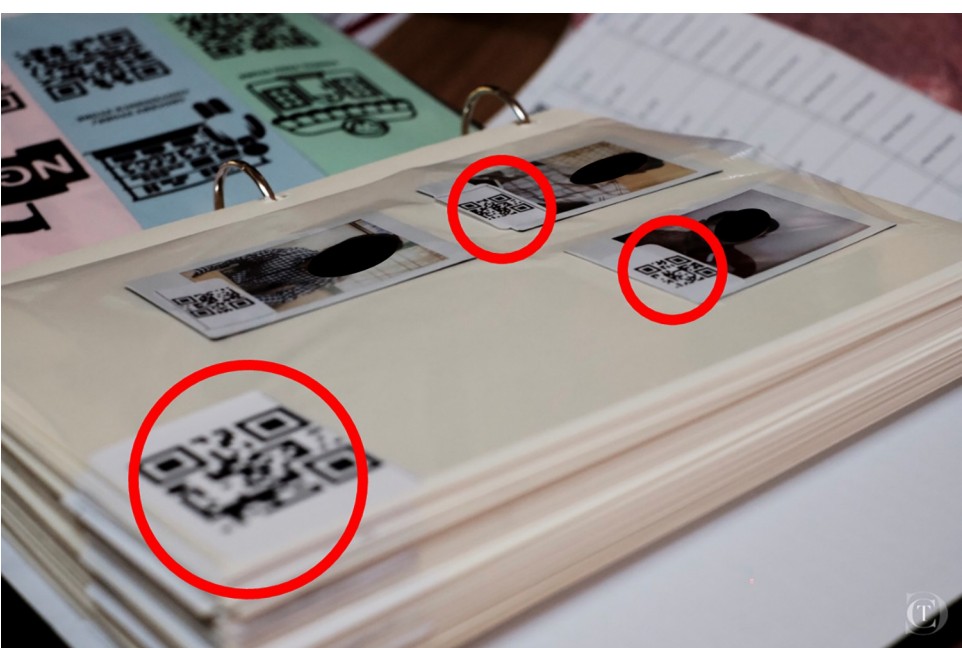

**Fig 2. Placement of participants' household and individual unique IDs on a photo album page.** Author Clarence Tam is the photographer for this figure.

using ODK Collect (v.1.21.0) software [7] on electronic tablets running the Android operating system.

## In-depth interviews

We then selected 28 key informants from participants' talking networks. These individuals were invited to take part in an in-depth interview to elicit information about the influence of social ties on their decision-making processes related to personal and animal health, and antibiotic use (S3 Appendix). Key informants represented nodes in the network that were well-connected (high in-degree, or receiving nominations from many different households), that served as bridges between different parts of the network (high betweenness) and isolated nodes in the network (low in-degree and low betweenness) (**Fig 3**). We also selected for in-depth interviews six individuals who were identified as sources of antibiotics in the two villages, including a health centre manager, village health animal worker, pharmacy owner, an animal feed store owner and two grocery store owners.

Interviews were conducted in Khmer by trained data collectors (CT, SH) using a semi-structured interview guide designed to elicit information about participants' antibiotic decision-making in infection prevention or treatment for themselves and their animals. To facilitate discussion about the importance of social relationships, we used a tiered semi-circular mat with coloured sectors of varying radius to represent increasing social distance from the interviewee (**Fig 4**). Interviewees were then asked to place photographs of individuals they had nominated in the network survey on the mat at different distances relative to themselves, representing how important nominees were as sources of information for their own health and for animal health matters. Interviews lasted 45 minutes on average and were audio-recorded with permission from the interviewee. Audio recordings were transcribed and translated into English verbatim for analysis.

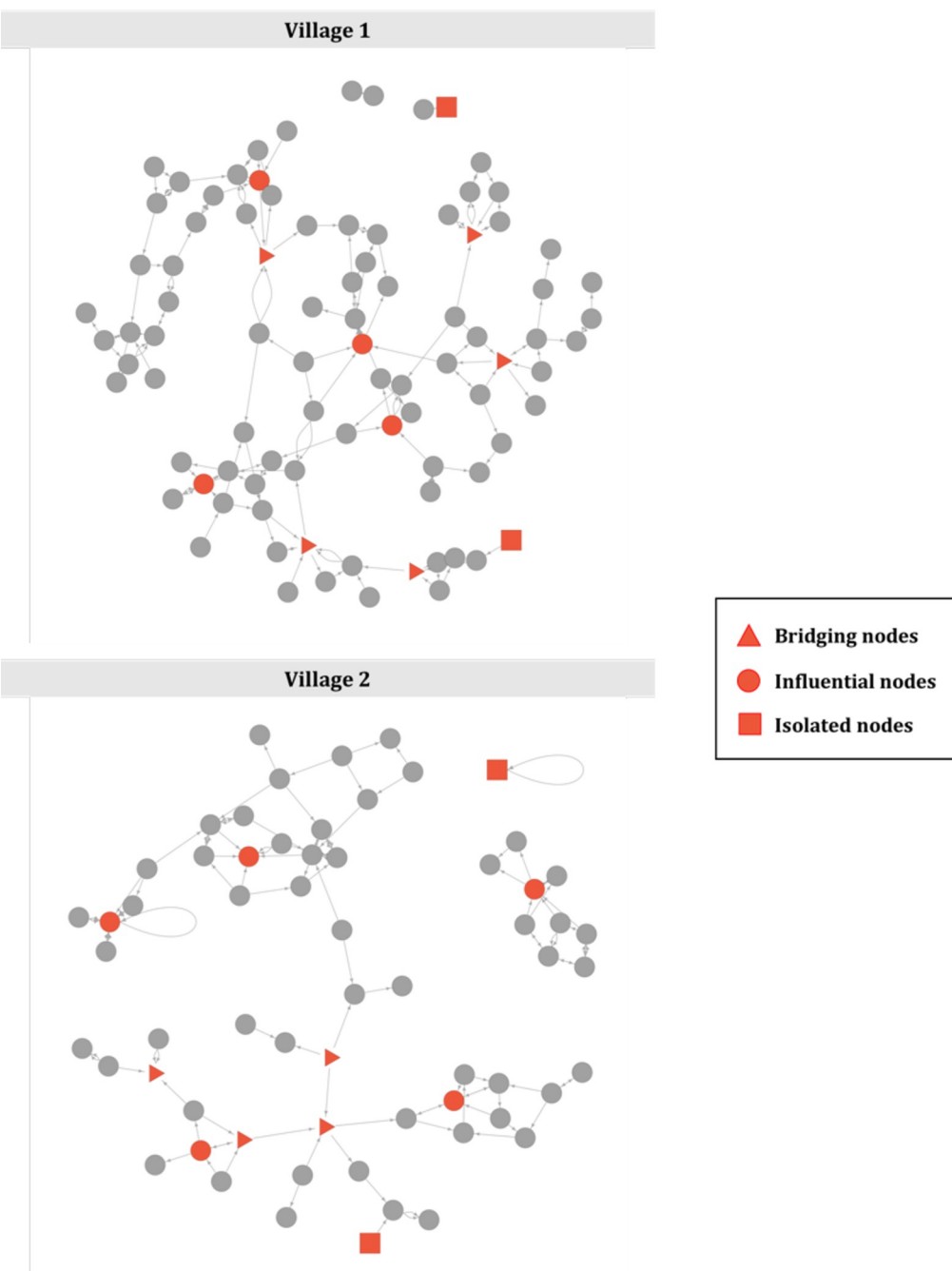

**Fig 3. Selection of key informant sources for in-depth interviews.** Nodes represent individual households. Key informants selected for in-depth interviews are shown in red.

## Data analysis

We first generated directed social networks to visualise the types of connections formed within each village. Next, we derived a score for each participant from ten variables that captured information about specific antibiotic use practices (S1 Table); higher scores indicated more favourable behaviours. Additionally, we assigned each participant a binary data string based on their responses to these ten questions. We then assessed the similarity between individuals'

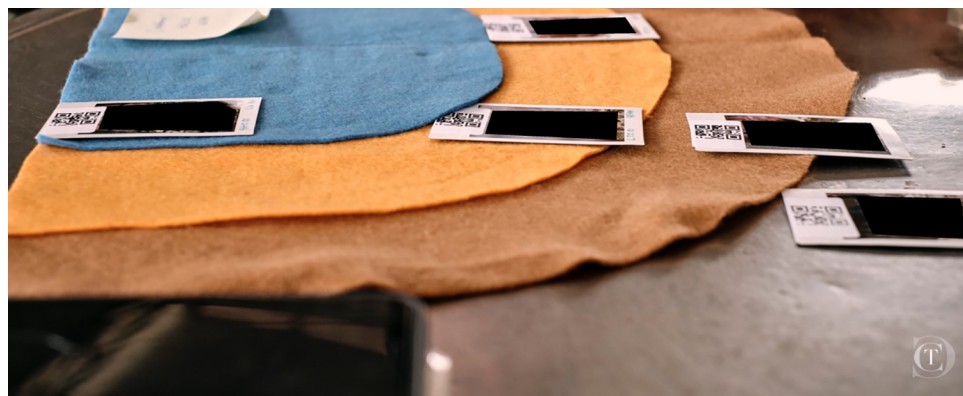

**Fig 4. Using an interactive activity, key informants place photographs of their nominated contacts on a mat at varying distances from themselves representing the strength of social connection across talking, work-related and health-related networks.** Author Clarence Tam is the photographer for this figure.

antibiotic practices by computing the pairwise matrix of Hamming distances for all dyads in the social network. The Hamming distance measured the number of positions at which these binary strings varied between two individuals.

We conducted univariable and multivariable linear (Gaussian) regression analyses to investigate the relationship between participants' antibiotic scores and their sociodemographic characteristics. We informed these findings with thematic analysis of qualitative data from key informant in-depth interviews, to provide contextual information about individuals' decision-making processes in relation to animal health and antibiotic use, and the influence of social ties on health behaviours.

We used exponential random graph modelling (ERGM) to evaluate the extent to which network parameters and covariates adequately represented the talking network structures observed in the study. This also allowed us to explore if ties were more likely to exist between certain nodes that shared similar attributes, such as antibiotic scores or low Hamming distances. Univariable analysis of each variable was conducted to select variables for the final ERGMs. Variables were selected based on a p-value cut-off point of 0.25.

Social network data were analysed using the igraph [8] (https://igraph.org) and statnet [9] (http://www.statnetproject.org) packages in R version 3.6.1 [10]. In-depth interview transcripts were imported into Dedoose Version 8.0.35 [11] to facilitate data coding, retrieval and analysis.

## Ethics statement

This study was approved by the institutional review board of the National University of Singapore (reference number: S-18-161) and the National Ethics Committee for Health Research, Cambodia (reference number: 203NECHR). Participants gave verbal consent to participate in each phase of the study, as well as permission for in-depth interviews to be audio-recorded. No participants' personal identifiers were audio recorded. Participant quotations are depicted by a study identifier to maintain anonymity. We returned the photographs to the individual participants once all data collection was completed.

## Results

### Participant and network characteristics

In villages 1 and 2, 143 and 105 participants from 97 and 67 households were included in the photographic census, while 89 and 56 participants completed the network survey respectively.

**Table 1. Baseline sociodemographic information for both villages.**

| Participant characteristics n (%) | Village 1 n = 143 | Village 2 n = 105 |
|---|---|---|
| **Sex** | | |
| Male | 48 (33.6) | 31 (29.5) |
| Female | 95 (66.4) | 74 (70.5) |
| **Age** | | |
| 21–29 | 18 (12.6) | 10 (9.5) |
| 30–39 | 36 (25.2) | 13 (12.4) |
| 40–49 | 13 (9.1) | 15 (14.3) |
| 50–59 | 37 (25.9) | 18 (17.2) |
| 60–69 | 22 (15.4) | 27 (25.7) |
| 70–79 | 12 (8.4) | 16 (15.2) |
| 80–89 | 5 (3.4) | 6 (5.7) |
| **Marital status** | | |
| Married or living with partner | 76 (53.1) | 78 (74.3) |
| Separated or divorced | 51 (35.7) | 21 (20.0) |
| Never married | 9 (6.3) | 6 (5.7) |
| Widowed | 7 (4.9) | 0 (0) |
| **Education** | | |
| No formal education | 31 (21.7) | 31 (29.5) |
| Up to primary education | 39 (27.3) | 35 (33.3) |
| Up to secondary education | 47 (32.8) | 18 (17.2) |
| More than secondary education | 26 (18.2) | 21 (20.0) |
| **Animal ownership**<sup>*</sup> | | |
| Cow | 80 (82.5) | 43 (64.2) |
| Chicken | 80 (82.5) | 43 (64.2) |
| Dog/Cat | 73 (75.3) | 39 (58.2) |
| Duck | 23 (23.7) | 20 (29.9) |
| Pig | 8 (8.2) | 2 (2.9) |
| Buffalo | 1 (1.0) | 0 (0) |

*only the head of each household was required to answer.

Most participants were female, married or living with a partner, and most commonly owned buffaloes and chickens (**Table 1**).

## Network information

In terms of who they spent the most time talking to on a typical day, participants from the network survey generally nominated family members and contacts from other households who were geographically close to them (**Fig 5**), most commonly citing convenience. Although individuals generally felt comfortable making casual conversation with others, they would not necessarily trust them with more sensitive information, especially if they were a non-familial tie:

> *"If just for normal talk and joking I choose this [contact, because] she is only for fun. . .I could not trust her" (HH 100, ID 250)*

Participants with identical antibiotic response patterns (Hamming distance = 0) had shorter geodesic distances in their talking networks (r = 0.66, p < 0.001) and lived closer to each other

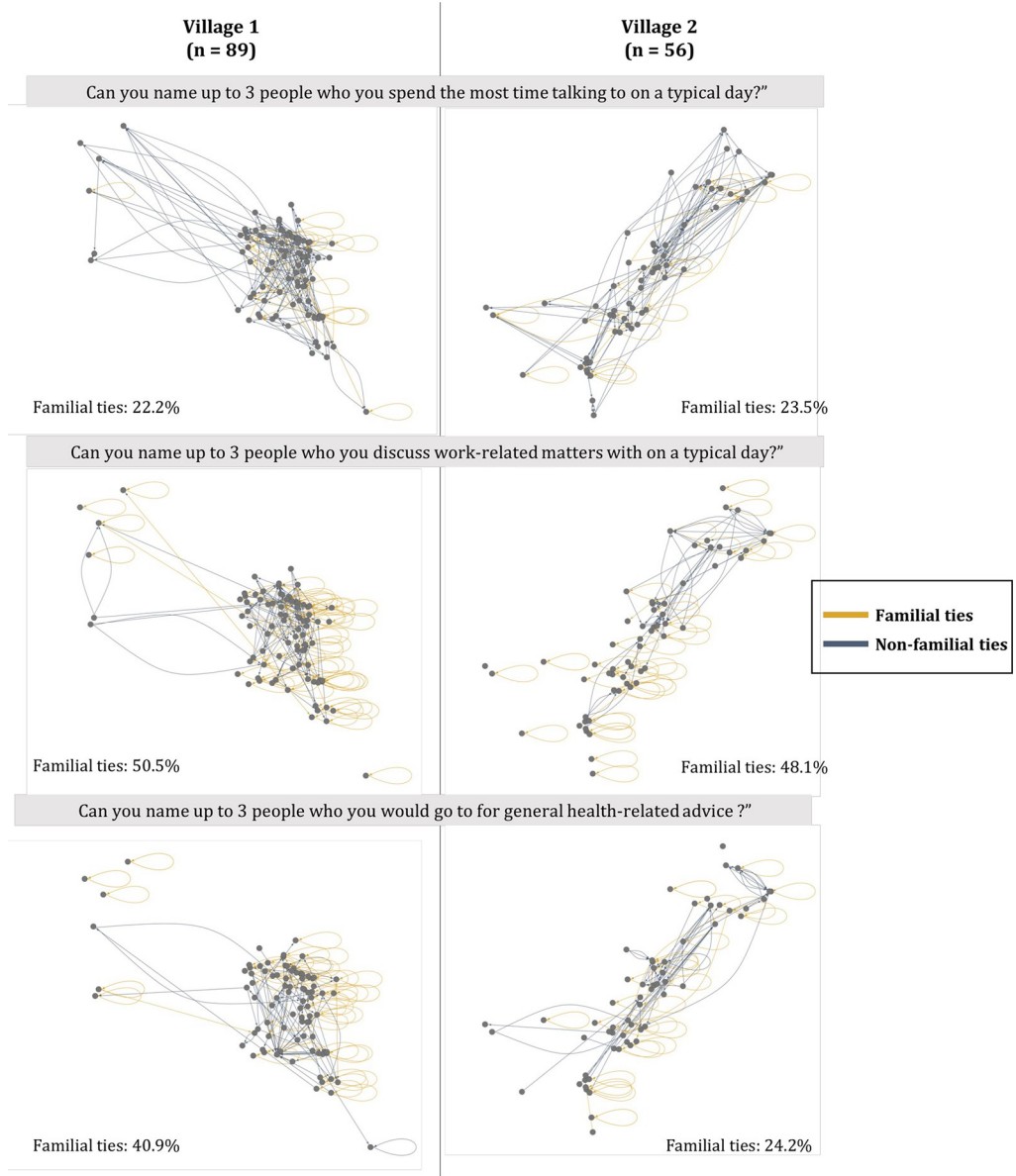

**Individual households are represented by nodes. Households are linked through nominations from village members, represented by directed edges. Edge colours differentiate connections between familial and non-familial ties, node sizes represent the number of times households were nominated, while loops indicate a within-household nomination.**

Fig 5. **Graphic representations of directed social networks observed in both villages.**

(r = 0.68, p = 0.02). Participants who lived closer to each other were also more likely to have shorter geodesic distances (r = 0.74, p = 0.02), indicating that social connections and antibiotic perceptions and practices in these communities correlate strongly with shorter geographic distance (**Fig 6**).

Work-related matters were also primarily discussed with familial connections or neighbouring households. In choosing whom to discuss work-related issues with, participants

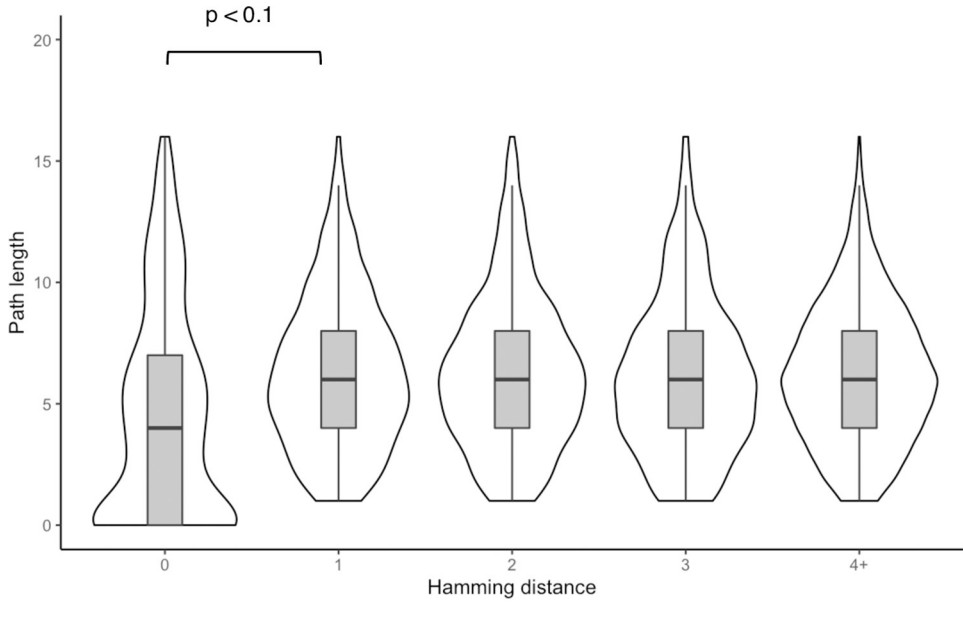

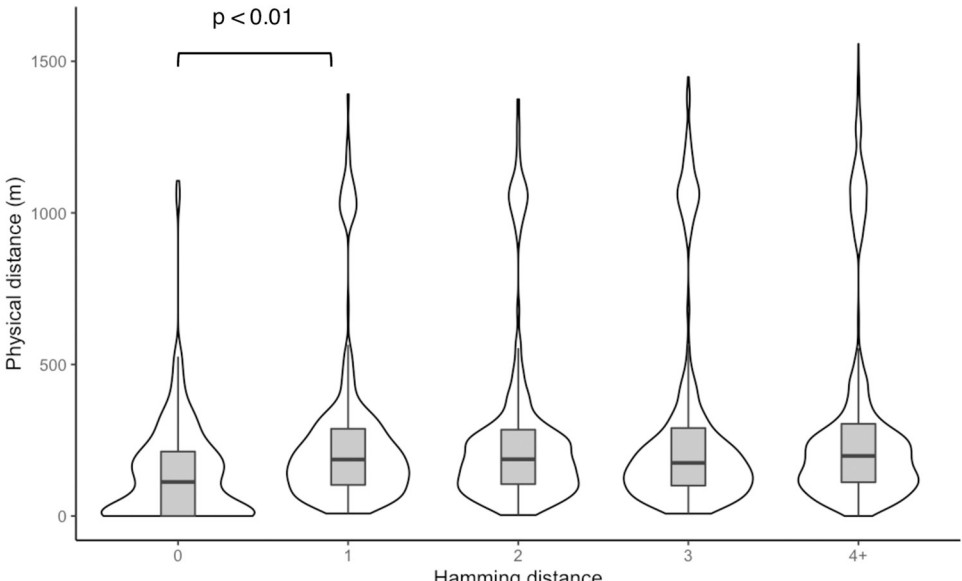

**Fig 6. Box plots of Hamming, geodesic and geographic distances.**

valued trustworthiness, availability, experience and convenience. Work-related issues discussed with others typically related to animal farming or agriculture.

> "I shared on seeds/seedling selection. . .and on local chicken breeds are more resistant than imported breeds, and vaccination, [and] high quality feed as my store also has commercial feed with high quality to sell to the clients. [I shared with] at least 10 people so far" (HH 73, ID 403)

In contrast, general health advice was typically sought from personnel at (government-run) health centres and (private sector) pharmacies. Those who turned to familial connections or

other villagers for health-related advice tended to do so if their family members had some healthcare training, had experienced a similar illness previously, or had good recommendations for a healthcare provider. Most participants were also willing to share this information with others:

> "Yes, I recommend them to go to see doctor who provide me services with satisfactions. . .after [I] have satisfied with that doctor, then [I] may tell to other villagers who get similar illness" (HH 96, ID 245)

## Antibiotic use behaviours

Key informant interviews identified pharmacies and health centres in nearby villages as the most common sources of antibiotics for human and animal use, as well as grocery stores in the village. Other sources of veterinary antibiotics identified in the survey included the local animal health worker (50%) and the animal feed store (27.4%).

In terms of human health, antibiotics were most commonly used to treat wounds (54.8% of survey respondents) and fever (53.4%), and less commonly for respiratory symptoms such as coughs (33.6%), sore throats (38.4%) and runny nose (24.0%). This was corroborated by key informant interviews:

> "I buy [antibiotics] and keep for use when there is wound that [is] caused by something sharp or pointed, or for my leg pain and high body temperature" (HH29, ID117)

When deciding whether to use antibiotics for their animals, participants often relied on their own knowledge (80.8%) or information from television, radio and the internet (41.1%). A quarter (25.9%) reported not using antibiotics to treat any animal diseases. Instead, they used alternative methods such as paracetamol or increased biosecurity practices such as improved hygiene.

When purchasing antibiotics, participants typically described their symptoms to the seller, asked for antibiotics to manage symptoms from a self-diagnosis, or asked for specific antibiotics by name. Participants also reporting feigning symptoms in order to obtain antibiotics from human health sources for use in animals:

> "If I buy medicines from the grocery store [there are] instructions on the packs so I can read a little bit, but the medicines from health centre. . .I don't tell them that [it is] for chicken, but I tell them I got sick, so they give me the medicines." (HH76, ID409)

Almost half of the participants (43.4%) had shared antibiotics with family and friends experiencing similar illness, but sharing antibiotics for unrelated symptoms was uncommon (4.1%). About a quarter (28.3%) said that they had shared their own antibiotics with their sick animals but were less likely to use human antibiotics for growth promotion (2.8%) or illness prevention (11.0%). A fifth (22.1%) and a third (33.1%) of participants usually got a prescription before buying antibiotics for themselves and their animals, respectively.

## Attitudes towards antibiotics

About a quarter (28.3%) of the participants in both villages said that they kept a supply of antibiotics at home whether they were sick or not; 18.6% of households surveyed were verified to have antibiotics based on photographs. Common types of leftover antibiotics found in participants' homes included amoxicillin, ampicillin, cephalosporin, tetracycline and lincomycin.

**Table 2. Multivariable analysis.**

|  | Coefficient | 95% CI |
|---|---|---|
| Estimate | 8.47*** | 7.11–10.35 |
| **Village** | | |
| Village 1# | | |
| Village 2 | -0.77** | -1.31–-0.23 |
| **Sex** | | |
| Male# | | |
| Female | 0.45 | -0.17–1.06 |
| **Age (years)** | -0.01 | -0.03–0.02 |
| **Education** | | |
| No education# | | |
| Primary | -0.52 | -1.49–0.45 |
| Secondary | -1.29** | -2.24–-0.33 |
| More than secondary | -1.48** | -2.53–-0.42 |
| **Marital status** | | |
| Married# | | |
| Never married | -0.25 | -1.22–0.71 |
| Separated/Divorced | -0.31 | -1.40–0.77 |
| Widowed | -0.46 | -1.11–0.20 |
| **Makes decisions about animals** | | |
| Myself# | | |
| Spouse | -0.77* | -1.40–-0.13 |
| Children | -1.29* | -2.40–-0.18 |
| Parent | -0.49 | -1.63–0.66 |
| Other | -0.73 | -2.04–0.57 |
| **Get health advice** | | |
| Not health professional# | | |
| Health professional | 0.14 | -0.38–0.67 |

#Reference group.

***p ≤ 0.001

**p ≤ 0.01

*p ≤ 0.05.

## Antibiotic scores and regression analyses

Participants' median antibiotic scores were 7.01 (range: 0–10). In multivariable regression analyses (**Table 2**) exploring the association between participants' sociodemographic characteristics and their antibiotic scores, we found that participants in village 2 had slightly lower antibiotic scores (β = -0.77; 95% CI: -1.31 –-0.23). When compared to participants with no formal education, participants with secondary (β = -1.29; 95% CI: -2.24 –-0.33) and more than secondary education (β = -1.48; 95% CI: -2.53 –-0.42) tended to have lower antibiotic scores. Additionally, compared to participants who make their own decisions about animal care, participants whose spouses (β = -0.77; 95% CI: -1.40 –-0.13) and children (β = -1.29; 95% CI: -2.40 –-0.18) made decisions about their household's animal care had lower antibiotic scores.

ERGM analysis of the talking network highlighted the importance of reciprocity and closed triads (**Table 3**), indicating that nominations in the network were likely to be reciprocated, and that participants who shared a social time in common were also more likely to have nominated each other (β: 1.04, 95% CI: 0.51–1.57; β = 0.99, 95% CI:0.64–1.34). In village 1, age

**Table 3. ERGM analyses.**

| | Talking network | | | | Work-related network | | | |
|---|---|---|---|---|---|---|---|---|
| | Village 1 | | Village 2 | | Village 1 | | Village 2 | |
| | Estimate | 95% CI | Estimate | 95% CI | Estimate | 95% CI | Estimate | 95% CI |
| **Network structural measures** | | | | | | | | |
| Intercept (Edges) | -5.34*** | -5.89 --4.79 | -6.00*** | -6.78 --5.22 | -6.01*** | -6.85 --5.17 | -4.89*** | -5.50 --4.28 |
| Reciprocity | 2.00** | 0.80–3.20 | 1.84*** | 1.04–2.64 | 1.87*** | 0.77–2.97 | - | - |
| Triad closure | 1.04*** | 0.51–1.57 | 0.99*** | 0.64–1.34 | 0.86** | 0.33–1.39 | 1.49** | 0.51–2.47 |
| **Homophily** | | | | | | | | |
| Age | 0.60** | 0.15–1.06 | 0.45* | 0.02–0.88 | 0.50* | 0.01–0.99 | 0.29 | -0.33–0.92 |
| Antibiotic scores | 0.26 | -0.21–0.73 | 0.45* | 0.02–0.88 | 0.46 | -0.01–0.93 | - | - |
| Sex | - | - | 0.47* | 0.08–0.86 | 0.26 | -0.23–0.75 | - | - |
| Family | - | - | 0.28 | -0.15–0.71 | 0.47 | -0.02–0.96 | - | - |
| *"I have shared antibiotics with my animals"* | 0.38 | -0.11–0.87 | 0.47 | -0.06–0.99 | 0.49 | -0.10–1.08 | 0.46 | -0.17–1.09 |
| *"I usually get a prescription before obtaining antibiotics for my animals"* | 0.43 | -0.08–0.98 | 0.84** | 0.29–1.39 | 0.85** | 0.26–1.44 | 0.30 | -0.25–0.85 |

homophily was a strong determinant of network structure–the log odds of a tie existing between two nodes were higher if they were closer in age (β: 0.60; 95% CI: 0.15–1.06). In village 2, network structure was strongly dependent on age and sex homophily. In addition, social ties were more likely to exist between individuals who had the same antibiotic scores (β: 0.45; 95% CI:0.02–0.88), or shared similar practices in relation to obtaining antibiotic prescriptions for their animals (β: 0.84; 95% CI:0.29–1.39).

Results from both villages are presented in coefficients on the log scale and their corresponding 95% confidence intervals. The intercept reflects the log odds of a tie existing between two participants if none of the other variables were included in the model. A negative effect indicates that talking ties were unlikely to be formed outside of the processes included in the model.

In the work-related networks, closed triads were important determinants of network structure in both villages (β: 0.86, 95% CI: 0.33–1.39; β = 1.49, 95% CI:0.51–2.47 respectively), while reciprocity was only observed in village 1 (β: 1.87, 95% CI: 0.77–2.97). Additionally, ties in village 1 were more likely to exist if participants were closer in age (β: 0.50, 95% CI: 0.01–0.99) or if they shared similar practices for obtaining animal antibiotic prescriptions (β: 0.85, 95% CI: 0.26–1.44).

## Discussion

In this mixed methods study, we investigated how relationships and environment can influence antibiotic behaviours in Cambodian rural farming communities. Our findings indicated that participants most commonly accessed antibiotics or learned animal antibiotic practices through village-level informal sources. While most participants reported not using antibiotics for animal growth promotion or illness prevention, common misconceptions surrounding both antibiotic effectiveness and resistance existed. Additionally, participants who lived closer or had shorter geodesic distances in the network were more likely to have similar overall antibiotic knowledge and attitudes, suggesting the importance of familial ties in the community as information tends to cluster within families.

In terms of overall antibiotic behaviours, we found that most of the participants used and accessed antibiotics similarly for themselves and their animals, concomitant with general

misconceptions they had about antibiotic effectiveness. For instance, antibiotics were inappropriately used in both humans and animals to treat inflammatory conditions such as wounds, pain and fever, while about a fifth of the participants had leftover antibiotics at home. Approximately a third of the participants also reported having shared antibiotics with their animals, while almost half of the participants said that they shared antibiotics with others with the same illness. The lack of clear guidelines in backyard farming drug use is not novel [12,13], and it continues to add to the selective pressure on bacterial populations in both clinical and agricultural sectors. This also has implications for food safety as well as the emergence and spread of new resistant pathogens from animals [14].

Factors that influenced participants' decisions to obtain antibiotics for themselves and their animals included perceived quality of medicines, pricing, convenience, recommendations from others as well as trust in the personnel selling antibiotics. Antibiotics were also often acquired without a prescription, commonly from sources with no formal healthcare training, such as untrained pharmacy attendants, village animal health workers, grocery and animal feed store owners. The substantial role of informal sources of healthcare has been mirrored in studies conducted in similar rural settings [4], emphasising the importance of engaging and training this sector to increase appropriate antibiotic practice.

While some participants knew about adverse effects of taking antibiotics inappropriately, a majority were unaware about the mechanisms and consequences of antibiotic resistance. Corresponding with other findings in similar settings [15–17], participants thought that resistance occurs when the body becomes resistant to antibiotics and had generally low levels of awareness about the clinical consequences of resistance. As alternatives to antibiotics for their animals, participants practiced good hygiene and improved farm cleanliness, both of which have potential to reduce inappropriate antibiotic use on farms [16,18]. Vaccines administered by the village animal health worker were also used for specific conditions in chickens and cattle.

Family members or neighbouring households were commonly nominated as important contacts across talking, work-related and health-related networks. We also found that trust in healthcare sources was especially important in who participants nominated for health-related information. Additionally, we found that participants who lived closer or had shorter geodesic distances in the network tended to work more closely together and had more similarity in their patterns of responses to antibiotic knowledge and attitude questions. The link between social connectedness and similarity in behaviours is well-studied in other contexts [19,20] and offers some insight into how antibiotic behaviours can spread through social contacts and influence [21,22]. Our findings indicate that information in these communities tends to travel short distances. This is supported by ERGM analysis, which showed that similarity in antibiotic perceptions and practices (as measured by the Hamming distance), geodesic and geographic distances did not have significant effects on the formation of ties after controlling for general network structures such as tie reciprocity and triad closure. This suggests close-knit communities in which the formation of social ties and dissemination of health information and behaviours is largely shaped by familial relationships and geographic proximity.

Despite the significant economic role that backyard farming has in many developing countries, research on animal antibiotic use has focused on commercial farms. The use of mixed methods in this research enabled greater understanding of antibiotic use in smallholder rural farming communities from varying perspectives and provided insights on shared antibiotic interactions in both humans and animals. A number of limitations should be borne in mind. The communities included in this study were chosen based on their previous involvement with CelAgrid's agricultural programmes. While this enabled us to establish rapport and trust with participating households, it may limit the generalisability of our results. Additionally, the language barrier between the research team and participants presented some methodological

challenges in cross-language qualitative interviews. To mitigate these challenges in ensuring trustworthiness and conceptual equivalence, extensive training was conducted with interpreters and the research team was present for all in-depth interviews. Further, information on antibiotic practices was self-reporting and we were unable to validate this through other means.

The findings from our study have implications for future public health interventions and regulations to address inappropriate antibiotic use. First, recognising that patterns of antibiotic use for human and animal health are similar, the development of multi-dimensional interventions is especially pertinent in communities with high concentrations of backyard farming. Potential points of intervention include the benefits of improved sanitation and waste management, reframing antibiotic effectiveness more specifically for conditions with bacterial causes, as well as the mechanisms and clinical consequences of antibiotic resistance. Additionally, with greater understanding of the substantial roles that informal health sources and health centres take on in rural communities not only for human health but also for animal health, future interventions should engage informal providers in healthcare training and delivery, providing the appropriate resources for adequate antibiotic provision. These interventions should be complemented by information disseminated via media sources, as well as influential nodes in the community that can help to encourage adoption of appropriate antibiotic behaviours through pathways of social influence. Lastly, future interventions in these communities should explore the potential roles of weak ties as well as brokers in health-related information.

## Conclusion

Data from varying perspectives are crucial in designing more effective interventions to reduce inappropriate antibiotic use. This study utilises mixed methods conducted in different stages to explore how relationships and environment can influence antibiotic behaviours in Cambodian rural farming communities. Across talking, working and health networks in both villages, we found that participants tended to nominate familial connections or neighbouring households who were geographically close to them. This also translated to close geodesic and geographical networks having similar overall antibiotic knowledge and attitudes. Results from this study also demonstrate the importance of engaging village-level informal sources in the provision of antibiotic-related information for both human and animal health, as well as in potential interventions to encourage appropriate antibiotic behaviours in lower-resourced settings.

## Supporting information

**S1 Table. Participant antibiotic scores.**
(PDF)

**S1 Appendix. Photographic census demographic questionnaire.**
(PDF)

**S2 Appendix. Network survey questionnaire.**
(PDF)

**S3 Appendix. In-depth interview topic guide.**
(PDF)

**S4 Appendix. Inclusivity in global research statement.**
(PDF)

## Author Contributions

**Conceptualization:** Jane Mingjie Lim, Borin Khieu, Li Yang Hsu, Clarence C. Tam.

**Formal analysis:** Jane Mingjie Lim.

**Funding acquisition:** Clarence C. Tam.

**Investigation:** Clarence C. Tam.

**Methodology:** Jane Mingjie Lim, Sokchea Huy, Ty Chhay, Clarence C. Tam.

**Project administration:** Jane Mingjie Lim, Sokchea Huy, Ty Chhay, Clarence C. Tam.

**Resources:** Li Yang Hsu.

**Supervision:** Borin Khieu, Clarence C. Tam.

**Visualization:** Jane Mingjie Lim.

**Writing – original draft:** Jane Mingjie Lim.

**Writing – review & editing:** Jane Mingjie Lim, Sokchea Huy, Li Yang Hsu, Clarence C. Tam.

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
