## [Decision Letter · Decision Letter 0]

23 Nov 2022

PGPH-D-22-01522

Understanding networks in rural Cambodian farming communities and how they influence antibiotic use

Dear Dr. Lim,

Thank you for submitting your manuscript to PLOS Global Public Health. After careful consideration, we feel that it has merit but does not fully meet PLOS Global Public Health’s publication criteria as it currently stands. Therefore, we invite you to submit a revised version of the manuscript that addresses the points raised during the review process.

Please submit your revised manuscript by . If you will need more time than this to complete your revisions, please reply to this message or contact the journal office at globalpubhealth@plos.org. Please include the following items when submitting your revised manuscript:

We look forward to receiving your revised manuscript.

Kind regards,

Ben Pascoe

Academic Editor

Journal Requirements:

1. Please include a complete copy of PLOS’ questionnaire on inclusivity in global research in your revised manuscript. Our policy for research in this area aims to improve transparency in the reporting of research performed outside of researchers’ own country or community. The policy applies to researchers who have travelled to a different country to conduct research, research with Indigenous populations or their lands, and research on cultural artefacts. The questionnaire can also be requested at the journal’s discretion for any other submissions, even if these conditions are not met.  Please find more information on the policy and a link to download a blank copy of the questionnaire here: https://journals.plos.org/globalpublichealth/s/best-practices-in-research-reporting. Please upload a completed version of your questionnaire as Supporting Information when you resubmit your manuscript. 2. Please amend your detailed Financial Disclosure statement. This is published with the article. It must therefore be completed in full sentences and contain the exact wording you wish to be published. a. State the initials, alongside each funding source, of each author to receive each grant.b. State what role the funders took in the study. If the funders had no role in your study, please state: “The funders had no role in study design, data collection and analysis, decision to publish, or preparation of the manuscript.”c. If any authors received a salary from any of your funders, please state which authors and which funders. If you did not receive any funding for this study, please simply state: “The authors received no specific funding for this work.”" 3. Please provide separate figure files in .tif or .eps format only and remove any figures embedded in your manuscript file. Please also ensure that all files are under our size limit of 10MB. For more information about figure files please see our guidelines: https://journals.plos.org/globalpublichealth/s/figures https://journals.plos.org/globalpublichealth/s/figures#loc-file-requirements 4. We have noticed that you have uploaded Supporting Information files, but you have not included a list of legends. Please add a full list of legends for your Supporting Information files after the references list. 
We have noticed that you have uploaded Supporting Information files, but you have not included a list of legends. Please add a full list of legends for your Supporting Information files after the references list.  5. Figures 2 & 4: Please confirm (a) that you are the photographer; or (b) provide written permission from the photographer to publish the photo(s) under our CC-BY 4.0 license. 6. Figure 1: please (a) provide a direct link to the base layer of the map (i.e., the country or region border shape) and ensure this is also included in the figure legend; and (b) provide a link to the terms of use / license information for the base layer image or shapefile. We cannot publish proprietary or copyrighted maps (e.g. Google Maps, Mapquest) and the terms of use for your map base layer must be compatible with our CC-BY 4.0 license.  Note: if you created the map in a software program like R or ArcGIS, please locate and indicate the source of the basemap shapefile onto which data has been plotted. If your map was obtained from a copyrighted source please amend the figure so that the base map used is from an openly available source. Alternatively, please provide explicit written permission from the copyright holder granting you the right to publish the material under our CC-BY 4.0 license. Please note that the following CC BY licenses are compatible with PLOS license: CC BY 4.0, CC BY 2.0 and CC BY 3.0, meanwhile such licenses as CC BY-ND 3.0 and others are not compatible due to additional restrictions.  If you are unsure whether you can use a map or not, please do reach out and we will be able to help you. The following websites are good examples of where you can source open access or public domain maps: * U.S. Geological Survey (USGS) - All maps are in the public domain. (http://www.usgs.gov) * PlaniGlobe - All maps are published under a Creative Commons license so please cite “PlaniGlobe, http://www.planiglobe.com, CC BY 2.0” in the image credit after the caption. (http://www.planiglobe.com/?lang=enl) * Natural Earth - All maps are public domain. (http://www.naturalearthdata.com/about/terms-of-use/)

Additional Editor Comments (if provided):

Your manuscript has now been assessed by two independent reviewers and while both reviewers see merit in the work, both raise concerns that would preclude publication in the current form. Both reviewers appreciated the mixed method approach used in the study, but I agree with reviewer #2 that many of the findings are overstated and not supported by the data collected and sections of the discussion will need to be revised in an updated version of the manuscript. Both reviewers also noted low numbers for some statistical tests that need to be caveated.

Reviewers' comments:

Reviewer's Responses to Questions

**Comments to the Author**

1. Does this manuscript meet PLOS Global Public Health’s publication criteria? Is the manuscript technically sound, and do the data support the conclusions? The manuscript must describe methodologically and ethically rigorous research with conclusions that are appropriately drawn based on the data presented.

Reviewer #1: Yes

Reviewer #2: Partly

2. Has the statistical analysis been performed appropriately and rigorously?

Reviewer #1: Yes

Reviewer #2: I don't know

3. Have the authors made all data underlying the findings in their manuscript fully available (please refer to the Data Availability Statement at the start of the manuscript PDF file)?

Reviewer #1: Yes

Reviewer #2: Yes

4. Is the manuscript presented in an intelligible fashion and written in standard English?

Reviewer #1: Yes

Reviewer #2: Yes

5. Review Comments to the Author

Reviewer #1: This interesting mixed-methods study explores factors influencing antibiotic use in humans and animals in rural Cambodia.

Strengths of the manuscript include:

- It is well written

- The methodology is sophisticated and well described

- The findings are an important contribution to the literature

Title:

- I would suggest adding the study type (mixed methods)

Methods:

- Some information on how the villages were selected would be interesting

- USD3 => USD 3 or 3 USD

- “No members of the public were involved in the study.” In the study design ?

Results:

- “and most commonly owned buffaloes and chickens” probably cows instead of buffaloes (only one household seemed to have owned a buffalo based on the table 1)

- In addition to percentages, it would be good to also always add numbers “n”

- Is any information available on how often participants misidentified other medicines as antibiotics

Reviewer #2: This is an interesting article that addresses the previously documented phenomenon of antibiotic use among Cambodian 'backyard' farmers. It uses a range of methodological approaches including household census, social network survey and in-depth interviews to document local knowledge about antibiotics and social informational ties between villagers. While the results are useful, it is unclear whether the authors really demonstrate 'how [networks] influence antibiotic use' as claimed in the title. A key question is the claim regarding the 'spread of antibiotic behaviours through social contacts and influence' as this survey does not constitute causal evidence. In addition, all data on antibiotic use is self-reported, and responses may well have been affected by the established close links between the livestock centre that facilitated the study and by the somewhat intrusive research approach involving taking photographs of individuals and sharing these as a means to measure social networks.

I am not qualified to comment on the statistical methods used to analyse the social network data which are highlighted in the paper, but it is unclear whey key informants were selected from participants' 'talking' networks rather than their 'health' networks, since the latter was the focus of the study. Although great emphasis is placed in the paper on the social network survey, it is not clear that this adds much to our knowledge beyond verifying the rather well known fact that in small rural communities, kinship and neighbourhood ties are of primary importance for information sharing and social relations.

In the discussion of limitations, it is acknowledged that all information on antibiotic practices is self-reported and this needs to be reflected accurately in the Results section; e.g. instead of asserting that 'participants often relied on their own knowledge...' or 'they used alternative methods...' (p.8) it should be explicitly stated that the participants reported relying on... reported using... etc. Likewise, where results are based on prompts such as the use of biosecurity practices, this should be explicitly noted in the results section, as this otherwise gives the impression that informants volunteered this information autonomously. The limitations section should also consider the potential effects of requiring informants to be photographed before participating in the research and of sharing photographs with other villagers.

In the Methods or Antibiotic scores section more details should be provided on how/why statements were selected for inclusion and the basis for developing the scoring system. The Results section is rather short and a number of findings not reported in this section are instead included in the Discussion, such as factors that influenced participants' decisions to obtain antibiotics, and ideas about anitibotic awareness (p.11). These results should be moved to the Results section and then discussed here.

In addition, some points included in the discussion are not based on any findings reported in the results section, such as the benefits of improved sanitation and waste management on AMR, so the authors sometimes make claims beyond the evidence presented.

6. PLOS authors have the option to publish the peer review history of their article (what does this mean?). If published, this will include your full peer review and any attached files.

**Do you want your identity to be public for this peer review?** For information about this choice, including consent withdrawal, please see our Privacy Policy.

Reviewer #1: No

Reviewer #2: No

---

## [Editor Report · Decision Letter 1]

17 Jan 2023

Understanding networks in rural Cambodian farming communities and how they influence antibiotic use: A mixed methods study

PGPH-D-22-01522R1

Dear Dr. Lim,

We are pleased to inform you that your manuscript 'Understanding networks in rural Cambodian farming communities and how they influence antibiotic use: A mixed methods study' has been provisionally accepted for publication in PLOS Global Public Health.

Best regards,

Ben Pascoe

Academic Editor

Thank you for addressing the reviewers concerns. Your article can now be published.